# Urinary proteomics reveals biological processes related to acute kidney injury in *Bothrops atrox* envenomings

**Lisele Maria Brasileiro-Martins**[1,2☺], **Sofia Angiole Cavalcante**[3☺], **Thaís Pinto Nascimento**[1,2,3], **Alexandre Vilhena Silva-Neto**[1,2], **Marlon Dias Mariano Santos**[4], **Amanda C. Camillo-Andrade**[4], **Juliana de Saldanha da Gama Fischer**[4], **Caroline Coelho Ferreira**[3], **Lucas Barbosa Oliveira**[3], **Marco Aurelio Sartim**[1,2,5], **Allyson Guimarães Costa**[1,2,6], **Manuela B. Pucca**[7], **Fan Hui Wen**[8], **Ana Maria Moura-da-Silva**[8], **Jacqueline Sachett**[1,8], **Paulo Costa Carvalho**[4], **Priscila Ferreira de Aquino**[3‡*], **Wuelton M. Monteiro**[1,2‡*]

1 Department of Research, Dr. Heitor Vieira Dourado Tropical Medicine Foundation, Manaus, Brazil, 2 School of Health Sciences, Amazonas State University, Manaus, Brazil, 3 Leonidas and Maria Deane Institute, Oswaldo Cruz Foundation, Manaus, Brazil, 4 Structural and Computational Proteomics Laboratory, Carlos Chagas Institute, Oswaldo Cruz Foundation, Curitiba, Brazil, 5 Department of Research, Nilton Lins University, Manaus, Brazil, 6 Nursing School, Amazonas Federal University, Manaus, Brazil, 7 Department of Clinical Analysis, School of Pharmaceutical Sciences, São Paulo State University, Araraquara, Brazil, 8 Immunopathology Laboratory, Butantan Institute, São Paulo, Brazil

☺ These authors contributed equally to this work.
‡ PFA and WMM also contributed equally to this work.
* priscila.aquino@fiocruz.com (PFA); wueltonmm@gmail.com (WMM)

**Data Availability Statement:** Spectral raw data is submitted to the PRIDE database under the code

## Abstract

Acute kidney injury (AKI) is a critical systemic complication caused by *Bothrops* envenoming, a neglected health problem in the Brazilian Amazon. Understanding the underlying mechanisms leading to AKI is crucial for effectively mitigating the burden of this complication. This study aimed to characterize the urinary protein profile of *Bothrops atrox* snakebite victims who developed AKI. We analyzed three groups of samples collected on admission: healthy subjects (controls, n = 10), snakebite victims who developed AKI (AKI, n = 10), and those who did not evolve to AKI (No-AKI, n = 10). Using liquid-chromatography tandem mass spectrometry, we identified and quantified (label-free) 1190 proteins. A panel of 65 proteins was identified exclusively in the urine of snakebite victims, with 32 exclusives to the AKI condition. Proteins more abundant or exclusive in AKI's urine were associated with acute phase response, endopeptidase inhibition, complement cascade, and inflammation. Notable proteins include serotransferrin, SERPINA-1, alpha-1B-glycoprotein, and NHL repeat-containing protein 3. Furthermore, evaluating previously reported biomarkers candidates for AKI and renal injury, we found retinol-binding protein, beta-2-microglobulin, cystatin-C, and hepcidin to be significant in cases of AKI induced by *Bothrops* envenoming. This work sheds light on physiological disturbances caused by *Bothrops* envenoming, highlighting potential biological processes contributing to AKI. Such insights may aid in better understanding and managing this life-threatening complication.

PXD041548. All other relevant data are in the manuscript and its supporting information files.

**Funding:** WMM was funded by Coordination for the Improvement of Higher-Level Education Personnel (CAPES), Brazil–Finance Code 001, Fundação de Amparo à Pesquisa do Estado do Amazonas (FAPEAM) (Pró-Estado Program-#002/2008, #007/2018, and #005/2019, and POSGRAD Program). Conselho Nacional de Desenvolvimento Científico e Tecnológico (CNPq) supported AMMS (n. 03958/2018-9), WMM (n. 309207/2020-7), and MBP (307184/2020-0) with productivity fellowships. The funders had no role in study design, data collection and analysis, decision to publish, or preparation of the manuscript.

**Competing interests:** The authors have declared that no competing interests exist.

## Author summary

Envenomings caused by *Bothrops* species are a public health problem in the Brazilian Amazon. These envenomings can lead to a renal clinical complication called acute kidney injury. A better understanding of the causes leading to this complication is crucial for patient care. We aimed to describe the protein profile in the urine of patients who suffered envenoming and developed acute kidney injury. For this, we used large-scale analysis methods to compare the urine protein content of three groups of samples: healthy individuals, and snakebite patients with and without acute kidney injury. We identified a total of 1190 proteins, 65 of which were exclusive to patients suffering from snake bite envenoming. For patients with a kidney injury outcome, 32 unique proteins were found, of which the most abundant were associated with the body's inflammatory response. We highlight 4 proteins, including serotransferrin and SERPINA-1. Furthermore, we also evaluated candidate biomarkers of kidney injury already reported in the literature, such as retinol-binding protein, cystatin-C, and hepcidin, concluding that they are significant in cases of kidney injury caused by a jararaca snakebite. This work can help to better understand this serious complication that threatens the lives of the population living in the Amazon.

## 1. Introduction

From the estimated 1.8–2.7 million people suffering from snakebite envenomings per year, nearly 81–138,000 die from complications, and about 400,000 live with long-lasting sequelae worldwide [1–3]. Brazil holds the 6th position in the number of snakebite envenomings worldwide, and ranks the 1st in South America, with 26–29.000 cases/year [4,5]. The snake species with the highest medical importance in Brazil and Latin America belong to the genera *Bothrops*, *Crotalus*, *Lachesis*, and *Micrurus* [6]. The species *Bothrops atrox* gains prominence in the Brazilian Amazon, causing 80–90% of snakebite accidents in this region [7]. *Bothrops* envenoming presents local and systemic symptoms triggered by the complex combination of metalloproteinases (SVMPs), serine proteases (SVSPs), phospholipases (PLAs), and other proteins present in the *Bothrops* venom [6,8–10]. The tissue damage at the bite site can be evidenced by edema, blistering, bleeding, and pain. The systemic manifestations of the venom include nausea, headache, mucosal hemorrhage, hematuria, hematemesis, disseminated intravascular coagulation, and shock [11,12]. Clinical and environmental factors can lead to complications such as necrosis, secondary infection, intracranial hemorrhage, compartment syndrome, amputation, and acute kidney injury (AKI) [11,13,14].

AKI is defined by a rapid increase in serum creatinine, accompanied by a decrease in urine volume [15]. It is a multifactorial disorder associated with chronic kidney disease (CKD) development and a higher risk of mortality [15]. AKI can impair the functional status of other organs, leading to short and long-term effects that contribute to patient morbidity and increased medical resource expenses [14–16]. Along with *Crotalus* and Russell's viper, *Bothrops* envenoming accounts for most snakebite-induced AKI cases reported worldwide [14,17,18]. In *Bothrops* snakebites, the registered incidence of AKI varies from 1.4 to 44.4% [13,14,17]. However, the real incidence of this complication is unknown due to the under-notification of snakebite accidents in rural areas and inconsistency in the criteria set for AKI in several studies [13,17]. The variability in AKI diagnosis criteria poses challenges for diagnosing and understanding the onset of this condition [19–21]. The Acute Kidney Injury Network (AKIN) proposed a serum creatinine increase of 50% or above 0,3 mg/dL in 48 hours to

diagnose AKI [21]. Despite its sensibility, the 48-hours frame hinders the potential for early interventions in emergency scenarios, such as snakebite envenomings, which can lead to delayed patient management [14,22].

The use of large-scale analysis to unravel potential biomarkers and pathway disturbances has shown promising results in various fields [19,20]. Among these, mass spectrometry-based proteomics is highlighted due to its versatility for different samples, sensitivity, and resolution power [23]. In contrast to genes, proteins are the main functional and structural cell components, thus, offering more well-tuned information about the changes in an organism/tissue condition [19,20,23]. Protein research has already prompted new insights in the context of renal lesions [19,20,22,24,25] as well as snake venom composition [26,27]. Nonetheless, the pathophysiological mechanisms of acute kidney injury are not well established so far, especially in the context of snakebite envenoming [14,17,22,24].

Therefore, we aimed to characterize the protein profile in the urine of *Bothrops* accident victims with clinical outcomes of AKI. We expect that the information provided in this study may enhance the clinical management of these patients, contributing to the development of alternative prevention and reduction strategies for severe kidney injuries in *Bothrops* envenoming victims.

## 2. Methods

### 2.1. Ethics statement

This study was conducted according to the guidelines of the Declaration of Helsinki and approved by the Ethics Committee of Fundação Medicina Tropical Dr. Heitor Vieira Dourado (CAEE: 31535420.1.0000.0005; protocol of approval n: 4.026.226). All participants signed a consent form after reading of the study objectives and procedures.

We performed a cross-sectional study analyzing a subset of patients (n = 20) victims of snakebite caused by the *Bothrops* genus. We recruited these patients from October 2019 to December 2020 at the Fundação de Medicina Tropical Doutor Heitor Vieira Dourado (FMT-HVD). FMT-HVD is the sole referral emergency hospital to treat snakebite accidents in Manaus (AM), a city in North Brazil located in the Amazon region.

### 2.2. Study design

For descriptive comparison matters, three groups of samples were submitted to proteomic analysis: healthy controls (n = 10), *Bothrops* snakebite victims that developed AKI (n = 10), and patients that did not develop this condition (No-AKI, n = 10).

The control group was conveniently sampled from healthy individuals with sex and age, pairing the snakebite victims' profiles. The inclusion criteria for snakebite patients were individuals of any sex or age, victims of *Bothrops* envenoming, without prior treatment, with at least two measures of serum creatinine, being the first before the antivenom and the second at least 48h after the first one. We excluded patients with diabetes mellitus, renal diseases, autoimmune disease, and continuously using non-steroidal anti-inflammatory drugs. We considered the accidents as caused by *Bothrops* through recognition of clinical symptoms of *Bothrops* envenoming or snake identification by a trained herpetologist when possible. As an additional confirming step, we performed a venom antigenemia assay consisting of an ELISA with Mab anti-*B. atrox* plus biotin.

The snakebite clinical severity was classified on hospital admission according to the Brazilian Ministry of Health Guidelines for Snakebite Diagnosis and Treatment, as mild cases, characterized by pain and/or local edema, affecting one or two body segments, may present coagulopathy, without distant bleeding; moderate cases, characterized by pain and evident

edema affecting three to four body segments, may present coagulopathy or other systemic symptoms, such as gingival bleeding, but without impairment of general condition, and severe cases, characterized by edema involving the entire affected limb (5 body segments), severe pain, may present coagulopathy or other systemic symptoms, such as hypotension, may also present distant bleeding, necrosis, and/or compartment syndrome.

AKI diagnosis and staging followed the Acute Kidney Injury Network (AKIN) criteria [21]. The patients who had a primary change in serum creatinine of $\geq$0.3mg/dL or $\geq$150 to 199% on their baseline value were considered to have AKI clinical outcomes. The stages of kidney injury according to AKIN [21] consider Stage 1 lesion—an increase in serum creatinine of $>$ 0.3mg/dL or $\geq$ 150 to 199% about baseline creatinine, Stage 2 lesion—an increase in 200 to 299%, and Stage 3 Lesion–an increase of $\geq$ 4.0mg/dL or $\geq$ 300% or patient starting Renal Replacement Therapy. To minimize data variability, only patients classified with AKI at Stage 2 lesion were selected for proteomic analysis.

**2.2.1. Data collection and demographics analysis.** To collect data regarding demographic and clinical characteristics, we applied interviews and consulted medical records.

**2.2.2. Sample collection and laboratory assays.** In total, we collected 15 ml of peripheral blood and 60 ml of urine samples from snakebite patients at hospital admission before antivenom. Parallel to regular hospital laboratory analysis, we analyzed creatinine, fibrinogen, and d-dimer levels. For fibrinogen and d-dimer analysis, the blood samples were collected in falcon tubes with 3,8% sodium citrate and 2% pentavalent antivenom as the internal protocol to neutralize the venom. For the determination of serum creatinine, the patients must have undergone two measurements—the first before the antivenom and the second at least 48 hours after the first one. For the measurement, we used the AA Kinetic Creatinine-liquid line (Wiener Laboratory, Santa Fe, Argentine). Using an ACL TOP 300 CTS coagulation analyzer (Werfen Instrumentation Laboratory, Barcelona, Spain), we measured fibrinogen by the Clauss method [28] and D-Dimer by immunoturbidimetry. The control group patients contributed 15 ml of urine and 5 ml of blood for serum creatinine analysis, performed as described for envenoming victims.

## 2.3. Proteomic sample preparation

**2.3.1. Protein extraction.** Two milliliters (2 ml) of snakebite patients' urine and two milliliters (2 ml) of healthy controls urine were thawed and centrifuged (3,000xg at 4˚C) for 50 min to precipitate cellular debris. To extract the proteins, we applied a methanol precipitation protocol followed by Urea/Thiourea (7M/2M) extraction. First, we added methanol 100% at nine times (9x) the supernatant volume, vortexed for 10 min, then incubated for 16h at -20˚C. After, we centrifuged (3,000xg at 4˚C) for 90 min. We added on the result pellet methanol 100% at five (5x) times the supernatant volume, manually homogenized for 10 min, then centrifuged (3,000) at 4˚C for 1h30 min. The resulting pellet dried at 4˚C overnight. Following, 150µl of urea/thiourea (7M/2M) solution was added to the precipitated material; it was vortexed for 1 min and incubated for 50 min on ice, then centrifuged (3,000xg at 4˚C) for 3 min. The protein content was quantified using the fluorometric test Qubit 3.0 Protein Assay Kit (Thermo-Fisher Scientific), as instructed by the manufacturer.

**2.3.2. Sample preparation for mass spectrometry.** One hundred micrograms (100 µg) of each sample were reduced with 10 mM of dithiothreitol (DTT) at 37˚C for 1h30 min, alkylated with 30 mM of iodoacetamide (IAA) at room temperature in the dark for 45 min, and then incubated with 5 mM of DTT for 15 min at room temperature. Lastly, the urea concentration was diluted to 1 M with ammonium bicarbonate (50 mM). Afterward, all samples were digested overnight with trypsin (Promega) at the ratio of 1/50 (w/w) (E/S) at 37˚C. A volume

of 10% trifluoroacetic acid (1% v/v final) was added to stop the enzymatic reaction. According to the manufacturer's instructions, the subsequent peptide mixture was quantified using the Qubit 3.0 (Thermo-Fisher Scientific). Each sample was desalted and concentrated using Stage Tips (STop and Go-Extraction TIPs) as previously described [29].

**2.3.3. LC-MS/MS analysis setting and data acquisition.** The desalted peptide mixture was resuspended in 0.1% formic acid and analyzed with an UltiMate 3000 Basic Automated System (Thermo Fisher) online with a Fusion Lumos Orbitrap mass spectrometer (Thermo Fisher). The mixture chromatographic separation occurred on a column (30 cm x 75 μm) packed in-house with ReproSil-Pur C18-AQ 1.9μm resin (Dr. Maisch GmbH HPLC) with a gradient of ACN (5–50%) in 0.1% formic acid for 125 min in a flow rate of 250 nL/min. The Fusion Lumos Orbitrap was set in data-dependent acquisition (DDA) mode and automatically alternated between full-scan and MS2 acquisition with a 60s dynamic exclusion list. The full scans range from 200 to 1500 m/z with 60,000 at m/z 200 resolution. The ten most intense ions captured in a 2s cycle time were selected for MS2, excluding those unassigned or with a 1 + charge state. The selected ions were isolated and fragmented using Higher-energy collisional dissociation with a normalized collision energy of 40. The fragment ions were analyzed with a resolution of 15,000 at 200 m/z. The mass spectrometer ionization settings were as follows: 2.5 kV spray voltage, no sheath or auxiliary gas flow, heated capillary temperature of 250˚C, predictive automatic gain control (AGC) enabled, and an S-lens RF level of 40%. Mass spectrometer scan functions and nLC solvent gradients were regulated using the Xcalibur 4.1 data system (Thermo Fisher).

**2.3.4. Search and identification of spectral data.** The data analysis for protein identification and relative quantitation was performed using the software PatternLab for Proteomics V (PLV), freely available at http://www.patternlabforproteomics.org [30]. For peptide/protein sequence identification, we applied a peptide spectrum match approach (PSM) using the search engine Comet 2021.01 rev. 0 [31]. The target-decoy database was prepared by including peptide sequences from *Homo sapiens* and *Bothrops atrox*, plus sequences from the 123 most common mass spectrometry contaminants for protein analysis, and a decoy database generated through reversion of each sequence from the target database [30]. The sequences from *H. sapiens* were downloaded on March 23rd, 2023, from the Swiss-Prot database. The *B. atrox* sequences were obtained from a transcriptome database kindly provided by collaborators at Butantan Institute. The search parameters for identification were configured at the High-High option [32], considering fully and semi-tryptic peptide candidates with mass ranging from 500 to 6,000 Da, up to two missed cleavages, 35 ppm for precursor mass, and fragment ion bins of 0.02 m/z for MS/MS. The amino acid residue modifications evaluated were carbamidomethylation of cysteine as fixed, oxidation of methionine, and carbamylation of lysine and arginine as variable.

**2.3.5. Validation of PSMs.** The Search Engine Processor (SEPro) was used to assess the validity of PSMs [31]. The identifications were grouped by charge state (2+ and ≥ 3+), and then by tryptic status, generating four distinct subgroups. The XCorr, DeltaCN, DeltaPPM, and Peaks Matches values were used to produce a Bayesian discriminator, in each group. The identifications were sorted in nondecreasing order according to the discriminator score. A false-discovery rate (FDR) of 2% at the peptide level based on the number of decoys was accepted for the cutoff score. This process was independently executed on each data subset, resulting in an FDR independent of charge state or tryptic status. Additionally, as filtering parameters, we required a protein score greater than three, and a minimum sequence length of six amino-acid residues. Lastly, identifications deviating by more than 10 ppm from the theoretical mass were discarded.

**2.3.6. Quantification and data analysis.** The quantitation was performed according to PatternLab's Normalized Ion Abundance Factors (NIAF) as a relative quantitation strategy

and as previously described [30] being NIAF the equivalent to NSAF [32], though applied to extracted ion chromatogram (XIC). Considering each patient sample, a biological replicate, we quantitated the 10 biological replicates with two technical replicates, independently, for each group condition.

The T-fold and the Approximately Area-Proportional Venn Diagram PLV's modules were used to describe and analyze the proteomic data between conditions. The TFold parameters were set as a minimum of 2 replicates, no normalized, 0.05 BH q-value, 0.66 F-Stringency, 0.0 L-Strigency, 15 upper fold cutoff, and 0.0001 p-value cutoff. The Venn Diagram was set on stringent mode, without counting reverse sequences and contaminants, a minimum of 2 biological replicates, and 0.05 probability. The results read in 'Total satisfying a minimum of 2' and 'Total' consider the proteins found in at least two samples within the condition analyzed–biological replicates. The results in 'Venn Sum' point out the number of proteins found in both replicates of a given sample, minus proteins found in one replicate of another sample. In this situation, the PLV's Venn Diagram dismisses both samples' identification, hence the difference between 'Total' and 'Venn Sum'. Then, we used online database tools such as the Reactome Pathway Analysis (www.reactome.org) [33] and the STRING software (ww.string-db.org) [34] to search for the biological pathways associated with and the interaction networks within the list of identified proteins.

## 3. Results

A total of 30 patients were included in the analysis, equally distributed among the three groups evaluated: healthy controls, *Bothrops* snakebite patients who developed acute kidney injury (AKI), and those who did not (No-AKI). A comprehensive summary of the demographic, clinical, and laboratory characteristics of the cohort can be found in Tables 1 and 2.

In overview, the population analyzed predominantly comprises men (90%), from rural areas (95%), with hypertension as the main comorbidity found among the envenoming victims (15% of patients). Furthermore, 20% of the individuals had experienced previous snakebites, and 55% suffered moderate accidents. Interestingly, despite 90% of the patients receiving medical treatment within less than 6 hours, 50% of this population developed AKI according to the AKIN criteria.

### 3.1. Urine protein profile

Our analysis identified up to 20,482 peptides and 2,168 proteins (with redundancy) with false discovery rates at the protein level under 2.31% for all search results. We were able to quantify 1,190 proteins with at least two unique peptides. The list of identified proteins is available in S1 File, tab S1. Notably, no *Bothrops* venom-derived protein or peptide was identified in the urine samples.

To identify proteins exclusive and shared between different conditions, we employed the "Venn Diagram" feature of the PLV software (Fig 1). The analysis revealed 436 proteins commonly found in all conditions, and 65 proteins exclusively detected in snakebite victims: 33 of these proteins were shared between AKI and No-AKI patients, while 32 were specific to the AKI samples. Notably, no proteins were identified as exclusive to the No-AKI group. The complete list of all proteins identified under each condition can be found in S2 File.

### 3.2. *Proteins exclusive to* B. atrox *snakebite victims' and shared between the different outcomes*

Using the Reactome Pathway Analysis tool (www.reactome.org) [33], we assessed the biological pathways associated with the 33 proteins identified in both clinical outcomes evaluated. Afterward, we performed a manual analysis of their relative abundance in each condition.

**Table 1. Demographic and clinical characteristics of *Bothrops* snakebite patients at admission.**

| Variable | Total (n = 20) | No-AKI (n = 10) | AKI* (n = 10) |
|---|---|---|---|
| **Age range (years)** | | | |
| 16–45 | 13 (65.0%) | 8 (80.0%) | 5 (50.0%) |
| 45–60 | 3 (15.0%) | 1 (10.0%) | 2 (20.0%) |
| >60 | 4 (20.0%) | 1 (10.0%) | 3 (30.0%) |
| **Sex** | | | |
| Male | 17 (85.0%) | 9 (90.0%) | 8 (80.0%) |
| **Occurrence zone** | | | |
| Rural | 19 (95.0%) | 10 (100.0%) | 9 (90.0%) |
| Urban | 1 (5.0%) | 0 (0.0%) | 1 (10.0%) |
| **Time to medical assistance** | | | |
| <6 hours | 18/20 (90.0%) | 9/10 (90.0%) | 9/10 (90.0%) |
| **Site of bite** | | | |
| Foot | 12/20 (60.0%) | 6/10 (60.0%) | 6/10 (60.0%) |
| Leg | 6/20 (30.0%) | 3/10 (30.0%) | 3/10 (30.0%) |
| Hand | 2/20 (10.0%) | 1/10 (10.0%) | 1/10 (10.0%) |
| **Previous snakebite** | 4/20 (20.0%) | 0/10 (0.0%) | 4/10 (40.0%) |
| **Use of tourniquet** | 2/20 (10.0%) | 1/10 (10.0%) | 1/10 (10.0%) |
| **Use of traditional medicines** | 5/20 (25.0%) | 1/10 (10.0%) | 4/10 (40.0%) |
| **Severity classification** | | | |
| Mild | 3/20 (15.0%) | 2/10 (20.0%) | 1/10 (10.0%) |
| Moderate | 11/20 (55.0%) | 4/10 (40.0%) | 7/10 (70.0%) |
| Severe | 6/20 (30.0%) | 4/10 (40.0%) | 2/10 (20.0%) |
| **Comorbidities** | 4/20 (20.0%) | 0/10 (0.0%) | 4/10 (40.0%) |
| Chronic gastric | 1/4 (25.0%) | 0/10 (0.0%) | 1/4 (25.0%) |
| Hypertension | 3/4 (75.0%) | 0/10 (0.0%) | 3/4 (75.0%) |
| **Concomitant medication** | | | |
| Amlodipine | 1/4 (25.0%) | NA | 1/4 (25.0%) |
| Losartan | 2/4 (50.0%) | NA | 2/4 (50.0%) |
| Omeprazole | 1/4 (25.0%) | NA | 1/4 (25.0%) |
| **Local manifestations** | 8/20 (40.0%) | 5/10 (50.0%) | 3/10 (30.0%) |
| Bleeding | 6/20 (30.0%) | 4/10 (40.0%) | 2/10 (20.0%) |
| Edema | 6/20 (30.0%) | 4/10 (40.0%) | 2/10 (20.0%) |
| Ecchymosis | 1/20 (5.0%) | 1/10 (10.0%) | 0/10 (0.0%) |
| Blisters | 2/20 (10.0%) | 0/10 (0.0%) | 2/10 (20.0%) |
| Necrosis | 1/20 (5.0%) | 1/10 (10.0%) | 0/10 (0.0%) |
| **Systemic manifestations** | 9/20 (45.0%) | 5/10 (50.0%) | 4/10 (40.0%) |
| Headache | 5/20 (25.0%) | 3/10 (30.0%) | 2/10 (20.0%) |
| Diarrhea | 1/20 (5.0%) | 0/10 (0.0%) | 1/10 (10.0%) |
| Vomit | 1/20 (5.0%) | 1/10 (10.0%) | 0/10 (0.0%) |
| Nausea | 2/20 (10.0%) | 0/10 (0.0%) | 2/10 (20.0%) |
| Gum bleeding | 2/20 (10.0%) | 2/10 (20.0%) | 0/10 (0.0%) |

NA–Not Applicable.

*All AKI patients were diagnosed at Stage 2 Lesion following AKIN criteria.

**Table 2. Laboratory characterization of the patients at admission.**

| Variable | Total (n = 20) | No-AKI (n = 10) | AKI (n = 10) |
|---|---|---|---|
| **Creatinine** | | | |
| Normal | 12/20 (60.0%) | 10/10 (100.0%) | 2/10 (20.0%) |
| High value | 8/20 (40.0%) | 0/10 (0.0%) | 8/10 (80.0%) |
| **Creatinine mean (SD)** | 1.2 (0.6) | 0.8 (0.2) | 1.5 (0.7) |
| **Urea** | | | |
| Normal | 16/20 (80.0%) | 10/10 (100.0%) | 6/10 (60.0%) |
| High value | 4/20 (20.0%) | 0/10 (0.0%) | 4/10 (40.0%) |
| **Urea mean (SD)** | 38.2 (19.1) | 28.8 (8.2) | 47.5 (22.6) |
| **Lactate dehydrogenase** | | | |
| Normal | 8/15 (53.3%) | 4/6 (66.7%) | 4/9 (44.4%) |
| High value | 7/15 (46.7%) | 2/6 (33.3%) | 5/9 (55.6%) |
| **Lactate dehydrogenase mean (SD)** | 600.4 (590.9) | 440.3 (152.7) | 707.1 (751.3) |
| **Creatine phosphokinase** | | | |
| Normal | 9/14 (64.3%) | 3/4 (75.0%) | 6/10 (60.0%) |
| High value | 5/14 (35.7%) | 1/4 (25.0%) | 4/10 (40.0%) |
| **Creatine phosphokinase mean (SD)** | 218.3 (199.4) | 134.8 (77.2) | 251.7 (226.0) |
| **Clotting time** | | | |
| Normal | 7/20 (35.0%) | 4/10 (40.0%) | 3/10 (30.0%) |
| Prolonged | 1/20 (5.0%) | 0/10 (0.0%) | 1/10 (10.0%) |
| Unclottable | 12/20 (60.0%) | 6/10 (60.0%) | 6/10 (60.0%) |
| **Prothrombin activity time** | | | |
| Normal | 3/19 (15.8%) | 2/9 (22.2%) | 1/10 (10.0%) |
| Prolonged | 11/19 (57.9%) | 5/9 (55.6%) | 6/10 (60.0%) |
| Unclottable | 5/19 (26.3%) | 2/9 (22.2%) | 3/10 (30.0%) |
| **Hemoglobin** | | | |
| Normal | 14/20 (70.0%) | 7/10 (70.0%) | 7/10 (70.0%) |
| High | 4/20 (20.0%) | 2/10 (20.0%) | 2/10 (20.0%) |
| Low | 2/20 (10.0%) | 1/10 (10.0%) | 1/10 (10.0%) |
| **Hemoglobin mean (SD)** | 14.2 (2.6) | 13.9 (3.3) | 14.5 (1.7) |
| **Leukocytes** | | | |
| Normal | 8/20 (40.0%) | 4/10 (40.0%) | 4/10 (40.0%) |
| High | 12/20 (60.0%) | 6/10 (60.0%) | 6/10 (60.0%) |
| **Leukocytes mean (SD)** | 11483.3 (4643.6) | 11602.3 (5958.0) | 11364.2 (3161.1) |
| **Platelets** | | | |
| Normal | 19/20 (95.0%) | 10/10 (100.0%) | 9/10 (90.0%) |
| Low | 1/20 (5.0%) | 0/10 (0.0%) | 1/10 (10.0%) |
| **Platelets mean (SD)** | 203538.0 (51744.2) | 218880.0 (33080.2) | 188196.0 (63522.1) |
| **Fibrinogen** | | | |
| Normal | 5/20 (25.0%) | 3/10 (30.0%) | 2/10 (20.0%) |
| High | 2/20 (10.0%) | 1/10 (10.0%) | 1/10 (10.0%) |
| Low | 13/20 (65.0%) | 6/10 (60.0%) | 7/10 (70.0%) |
| **D-dimer** | | | |
| Normal | 2/19 (10.5%) | 2/9 (22.2%) | 0/10 (0.0%) |
| High | 17/19 (89.5%) | 7/9 (77.8%) | 10/10 (100.0%) |
| **Urine hemoglobin** | 5/10 (50.0%) | 0/2 (0.0%) | 5/8 (62.5%) |
| **Urine leukocytes** | 3/10 (30.0%) | 1/2 (50.0%) | 2/8 (25.0%) |
| **Urine crystals** | 1/10 (10.0%) | 0/2 (0.0%) | 1/8 (12.5%) |

*(Continued)*

**Table 2.** (*Continued*)

| Variable | Total (n = 20) | No-AKI (n = 10) | AKI (n = 10) |
|---|---|---|---|
| Urine casts | 6/10 (60.0%) | ½ (50.0%) | 5/8 (62.5%) |

**Reference values**: Lee-White clotting time: ≤ 9 min; Prothrombin activity time: 13 secs; Creatinine: 0.5 ± 1.2 mg/dL; Lactate dehydrogenase: 211 ± 423 IU/L; Creatine phosphokinase: 24 ± 190 IU/L; Leukocytes: 4,000 ± 10,000/mm3; Fibrinogen: 180 ± 350 mg/dL; D-dimer: < 500 ng/dL. Hemoglobin: 12.5±15.5 g/dL; Leukocytes: 4,000 ±10,000/mm$^3$; Platelets: 150,000±450,000/mm$^3$; Fibrinogen: 180±350 mg/dL; D-dimer: <500ng/dL.

*All AKI patients were diagnosed at Stage 2 Lesion following AKIN criteria.

As a result, Fig 2 lists 10 significant pathways associated with the proteins commonly found between AKI and No-AKI urine, and Fig 3 displays the 11 most abundant proteins in this comparison and their respective relative abundance in each condition.

The most abundant proteins found in the urine of No-AKI patients were myoglobin (MB), carboxypeptidase B2 (CPB2), and coagulation factor IX (F9). The main pathways linked to these proteins were 'Formation of Fibrin Clot', 'Defective factor VII causes hemophilia A', 'Defective factor IX causes thrombophilia', and 'Intracellular oxygen transport'.

Meanwhile, urine proteins more abundant in the AKI group include Apolipoprotein C-II (APOC2), Dermicidin (DCD), Complement components C8 gamma chain (C8G) and C2, insulin-like growth factor binding protein 6 (IGFBP6), SH3 domain-binding glutamic acid-rich-like protein (SH3BGRL), osteoglycin (OGN), and serglycin (SRGN). The main pathways associated with these proteins were the 'CLEC7A/inflammasome pathway', 'Terminal pathway of complement', and 'AIM2 inflammasome' [33,35–37]. Moreover, the pathways 'Platelet degranulation', 'Regulation of complement cascade', and 'Peptide hormone metabolism' were found to be associated with proteins abundant in both clinical outcomes.

**3.2.1. Proteins exclusively found in snakebite victims developing AKI.** Using the STRING software, we explored potential interactions within the exclusive protein set of AKI patients, aiming to discover the biological processes associated with these proteins. As a result, we found a network with 10 proteins with at least 3 points of interaction (Fig 4).

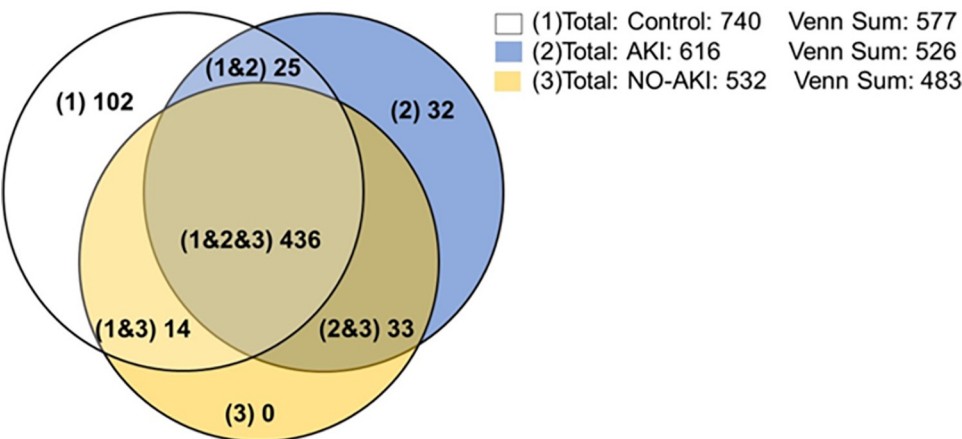

**Fig 1. Venn Diagram showing the common and exclusive proteins identified in the urine of *B. atrox* snakebite victims.**

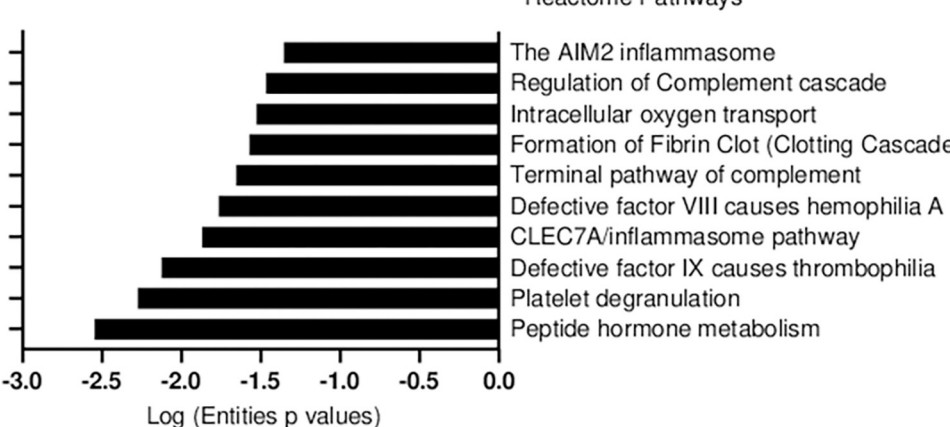

**Fig 2. Biological pathways of proteins exclusively found in the urine of *Bothrops* snakebite victims, shared by both clinical outcomes.**

Regarding Fig 4, the green-marked proteins were assigned as part of a network comprising elements from the complement cascade, the coagulation cascade, and lipid metabolism that may interact, such as the complement 8 beta chain (C8B) and alpha chain (C8A), complement 5 (C5) and Serum paraoxonase/arylesterase 1 (PON1). The term 'acute phase response' refers

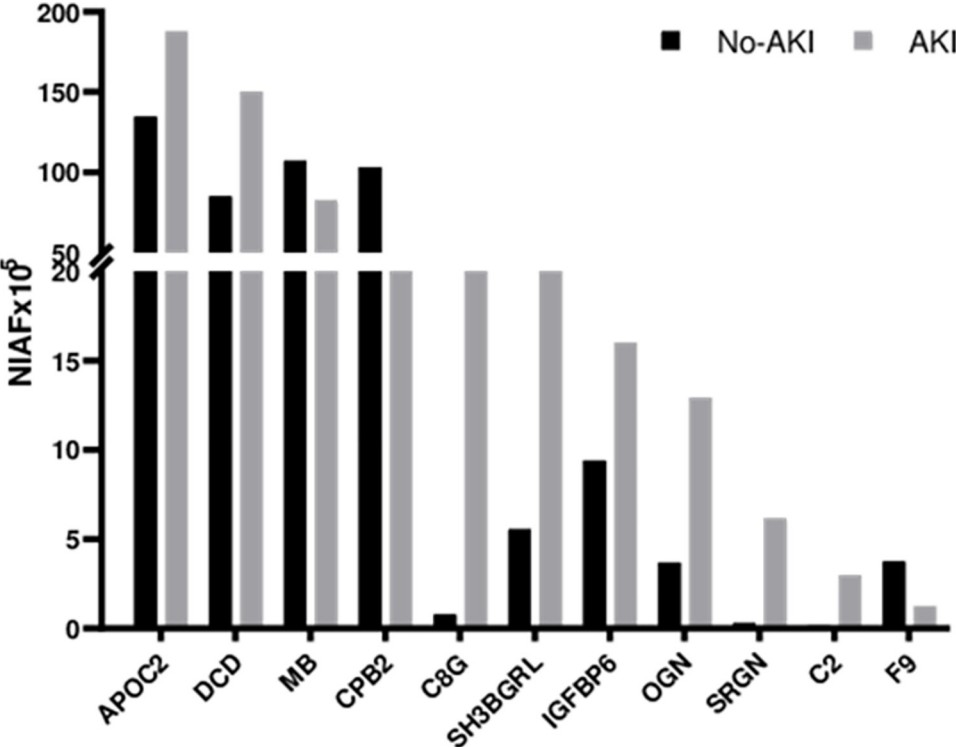

**Fig 3. Distribution of the 10 most abundant proteins exclusive to *Bothrops* snakebite victims' urine between No-AKI and AKI patients.** Protein listed: Apolipoprotein C-II (APOC2), Myoglobin (MB), Carboxypeptidase (B2CPB2), Dermcidin (DCD), Component C8 gamma chain (C8G), SH3 domain-binding glutamic acid-rich-like protein (SH3BGRL), Insulin-like growth factor binding protein 6 (IGFBP6), Osteoglycin (OGN), Serglycin (SRGN), Complement C2 (C2), and Coagulation factor IX (F9).

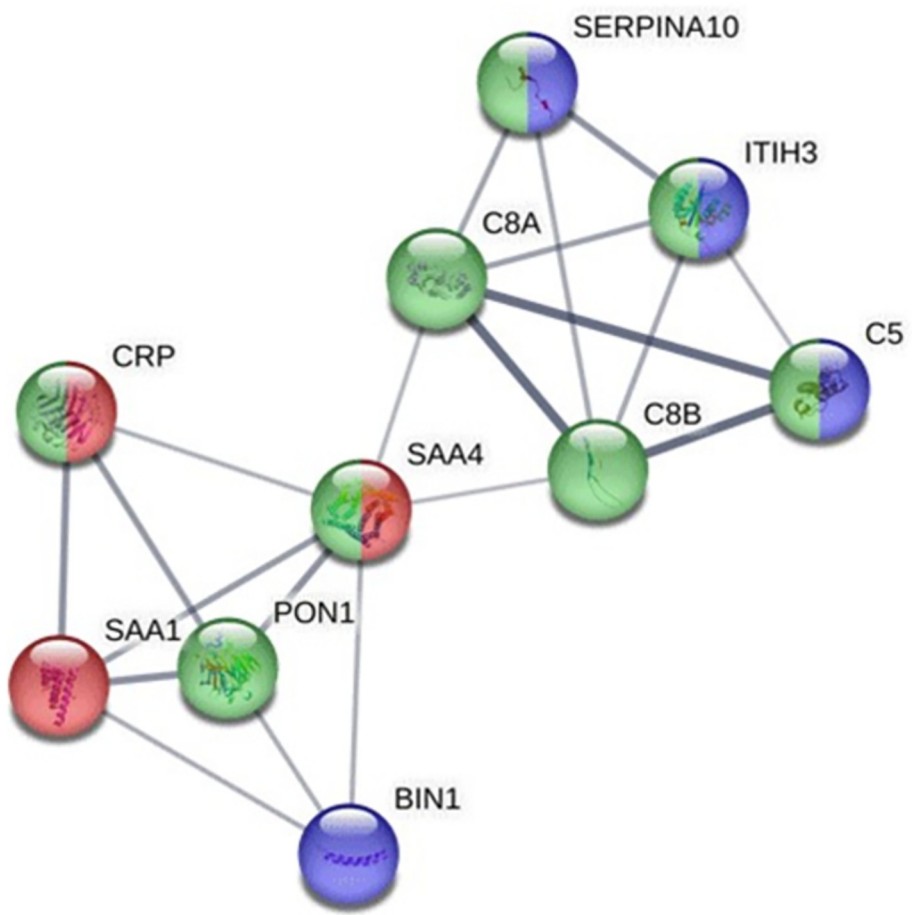

**Fig 4. Protein-protein interactions within the group of proteins exclusively found in the urine of Bothrops snakebite patients with acute kidney injury.** Colors represent the associated biological processes: Red–Acute phase response (GO:0006953); Green–Mixed processes, including complement and coagulation cascades, and lipoprotein particles (CL:19706); Blue–Endopeptidase inhibitor activity (GO:00048630).

to the acute inflammatory response involving non-antibody proteins whose plasma concentrations increase or decrease by at least 25% in response to an infection or injury; marked with red, the proteins in this process include C-reactive protein (CRP) and the Serum Amyloid A proteins—SAA1 and SAA4 [38–41]. The blue-marked proteins point out components associated with the 'endopeptidase inhibitor activity process': Protein Z-dependent protease inhibitor (SERPINA10), Myc box-dependent-interacting protein 1 (BIN1 and inter-alpha-trypsin inhibitor heavy chain H3 (ITIH3).

**3.2.2. Differential abundance in proteins common to all conditions.** To unravel the differential distribution of proteins identified and quantified among all the study conditions, we used the 'TFold' module on PLV. By comparing the findings among the different groups of snakebite victims, we obtained the results shown in Fig 5. Additionally, the proteins and their respective fold-changes can be found in Online Source 2.

As a result, we observed that serotransferrin (TF), alpha-trypsin 1 (SERPINA1), Alpha-1B-glycoprotein (A1BG), and NHL repeat-containing protein 3 (NHLRC3) exhibited differential abundance in the AKI patients' group.

### 3.3. Identification of biomarkers candidates for kidney injury

In our dataset, we carefully examined the identification of previously described candidates for acute kidney injury or renal injury, whether provoked by snakebite venom or not. Our search for previously described candidates included terms such as kidney injury, acute kidney injury, acute renal injury, renal biomarkers, proteomic snakebite, and envenoming into Google Scholar and PubMed search engines. We aimed to assess their potential relevance in the evaluation of *Bothrops* accidents. Fig 6 shows the distribution of protein abundance between No-AKI and AKI samples for 13 biomarkers that have been suggested as relevant in this context.

Our data suggest retinol-binding protein (RBP4), beta-2-microglobulin (B2M), cystatin-C (CST3), hepcidin (HAMP), and fatty acid-binding protein (L-FABP) hold promise as candidates for kidney damage surveillance in *Bothrops* envenoming. However, neutrophil gelatinase-associated lipocalin (NGAL), Clusterin (CLU), and alpha-2-HS-glycoprotein (AHSG) did not show potential as prognostic parameters in *Bothrops* snakebite-induced AKI based on our data. Interestingly, we found an abundance of protein S100-A8 (S100A8) and protein S100-A9 (S100A9) in No-AKI patients' urine.

In addition to the mentioned markers, other appointed indicators of kidney injury, such as kininogen-1 (KNG1), aminopeptidase-N (ANPEP), and glutathione S-transferase P (GTSP) [42–49], were found in our data (S1 File, tab S1); however, they did not present a significant abundance or difference among the studied groups.

## 4. Discussion

Multiple factors may influence the onset of acute kidney injury in *Bothrops* envenoming including sex, age, presence of comorbidities (i.e., diabetes mellitus and hypertension), delay

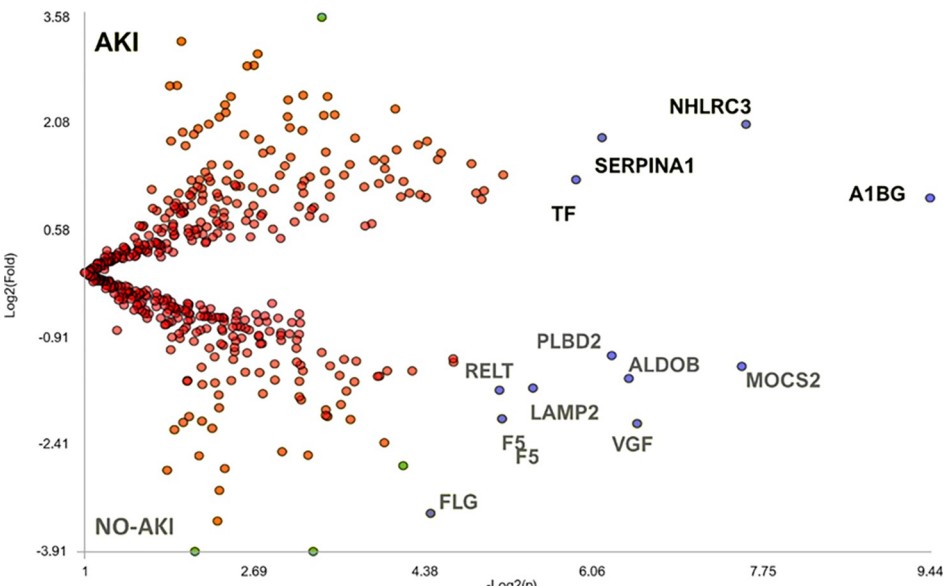

**Fig 5. Differential abundance of proteins in the urine of *Bothrops* snakebite victims with distinct outcomes regarding acute kidney injury.** Each protein was represented in the graph as a dot, plotted based on the (-"Log")2(" p-value) on the x-axis and Log2 (Fold change) on the y-axis. Red dots indicate proteins that did not meet the q-value cutoff or the fold change cutoff. Green dots represent proteins that met the fold change but not the q-value. Blue dots signify proteins that satisfied both fold change and q-value (0.05). The proteins plotted include alpha-1B-glycoprotein (A1BG), NHL repeat-containing protein 3 (NHLRC3), serotransferrin (TF), alpha-trypsin 1 (SERPINA1), molybdopterin synthase sulfur carrier subunit (MOCS2), fructose-bisphosphate aldolase B (ALDOB), neurosecretory protein VGF (VGF), putative phospholipase B-like 2 (PLBD2), lysosome-associated membrane glycoprotein 2 (LAMP2), tumor necrosis factor receptor superfamily member 19L (RELT), coagulation factor V (F5) and filaggrin (FLG).

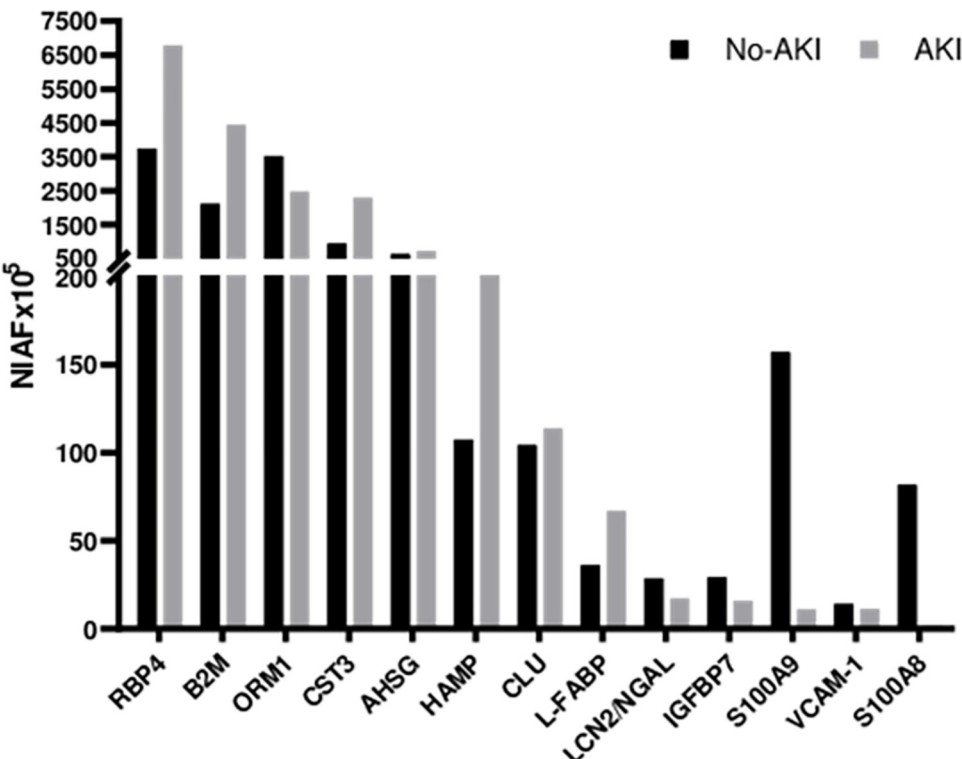

**Fig 6. Distribution of literature-appointed biomarkers for kidney injury in *Bothrops* snakebite urine samples.** The proteins listed are retinol-binding protein (RBP4), beta-2-microglobulin (B2M), alpha-1-acid glycoprotein 1 (ORM1), cystatin-C (CST3), alpha-2-HS-glycoprotein (AHSG), hepcidin (HAMP), clusterin (CLU) and fatty acid-binding protein (L-FABP), neutrophil gelatinase-associated lipocalin (NGAL), insulin-like growth factor-binding protein 7 (IGFBP7), protein S100-A9 (S100A9), vascular cell adhesion protein 1 (VCAM1) and protein S100-A8 (S100A8).

in reaching medical assistance, accident severity, snake species and size, snake geographic distribution, and amount of venom inoculated [8,11–13, 50–52]. The main mechanism attributed to inducing AKI involves inflammation, formation of immune complexes, hemodynamic disorders, and direct action of venom on kidney tissues [13,14,18,24,53–55]. Hematological disorders are frequently associated with snakebite-induced AKI; in this context, these disorders encompass hemoglobinuria, disseminated intravascular coagulopathy, thrombotic microangiopathy, and glomerular microthrombi deposition [14,17,18,43]. In a minor proportion, hypertension and myoglobinuria might provoke kidney injury [14,18,24,56]. These events predominantly lead to a decrease in renal vascular resistance (RVR), glomerular filtration rate (GFR), and urine flow [17,18,43]. However, it is noteworthy that the description of these mechanisms for *Bothrops* envenoming primarily comes from studies conducted in animal models and cell cultures [17,55,57,58]. In contrast, the data presented in this study originates from the urine analysis of *Bothrops* snakebite patients.

The urine proteome of the snakebite envenomed patients revealed an abundance of classical protein biomarkers of renal injury, such as albumin, along with more recently identified candidates for kidney injury, such as retinol-binding protein 4 (RBP4), beta-2-microglobulin (B2M), and cystatin-C (CST3) [42,43,49]. In our comprehensive analysis, we identified a group of 65 proteins exclusively found in the urine of snakebite victims. By examining 32 proteins shared between both AKI and No-AKI patients, we observed a group of proteins with higher levels in the urine of AKI; Searching these through a pathway database analysis, we found these were associated with lipid metabolism and inflammation processes, particularly

involving the AIM2 and CLEC7A (Dectin-1) inflammasome pathways and the complement cascade. Inflammation plays a significant role in both *Bothrops'* envenoming and the onset of AKI [58–61]. The venom's direct action triggers inflammation while the complement cascade process further debilitates the patient [55,62,63]. The precise venom components and pathways responsible for this condition remain a subject of investigation. Our data suggest the AIM2 and Dectin-1 inflammasome pathways, along with complement cascade elements C2 and C8, and lipidic disorders associated with apolipoprotein C-II (APOC2), can be associated with the development of renal failure in *Bothrops* snakebites.

Regarding the protein profile found in No-AKI patients, we highlight the great abundance of myoglobin and carboxypeptidase B2 (CPB2). The elevated myoglobin levels in this group add a layer of evidence of myoglobinuria as a less contributing factor to kidney injury in *Bothrops* envenoming, particularly *B. atrox* [14,18,24,56]. This could be attributed to differences in myotoxic venom components among *Bothrops* species. For instance, the venom of *B. atrox* has been reported to contain 5 to 8-fold lower myotoxic activity compared to venom from species such as *B. jararacussu*, *B. moojeni*, *B. neuwiedi*, and *B. pradoi* [64]. Furthermore, carboxypeptidase B2 (CPB2), also known as a thrombin-activatable fibrinolysis inhibitor (TAFI), plays a role in down-regulating fibrinolysis [57,65]. Previous studies have demonstrated that TAFI presents an important role in preventing tubulointerstitial fibrosis in obstructive nephropathy [66]. In this context, while it increases the risk of venous thrombosis, CPB2 may also act to mitigate direct kidney damage caused by fibrinolysis byproducts [65].

It is important to highlight the identification of 32 proteins exclusively found in the AKI patient group. Within this group, we observed a core network of proteins associated with the acute phase response, the complement and coagulation cascades, lipid metabolism, and endopeptidase inhibitor activity. These findings strongly support the idea of snakebite envenoming inducing a typical acute response, resembling the body's reaction to acute traumas [17]. It also reinforces the potential role of the complement cascade and lipidic disorders in inflammation feedback during snakebite envenoming [13,14,18,24,55,57]. Further, the most differentially abundant proteins found to be associated with renal failure were serotransferrin (TF), alpha-trypsin 1 (SERPINA1), alpha-1B-glycoprotein (A1BG), and NHL repeat-containing protein 3 (NHLRC3). serotransferrin or transferrin, is the main glycoprotein responsible for the blood transport of iron. It carries the iron from sites of absorption and heme degradation to sites of storage and utilization [67]. High levels of urine transferrin were reported on a cisplatin-induced AKI murine model investigating persistent renal vulnerability after AKI [68]. Additionally, it also has been found in patients' blister fluids following envenoming by snakes in India, which could be associated with local tissue damage [8,9,69,70]. The high levels of serotransferrin in the urine of AKI patients might indicate iron toxicity as an early and preeminent event in the establishment of AKI, like observations in chronic kidney disease and renal injuries promoted by other diseases [67]. Regarding the abundance of SERPINA1, a serine protease inhibitor, we hypothesize it may serve as a direct response to the serine proteases in *Bothrops* venom, which can directly affect the kidneys [8,9,18,55–57,69]. Although high levels of SERPINA1 have been found in acute kidney injury patients and nephrotic syndromes, such as glomerulonephritis [71,72], this protein is also associated with a protective role in maintaining kidney membrane integrity [73]. Therefore, its presence in urine might signal glomerular membrane impairment, and its abundance could correlate to the venom-inoculated dosage and toxicity [72,73]. Alpha-1B-glycoprotein (A1BG) is a plasma glycoprotein of unknown function; transcriptome analysis shows enrichment in liver tissue and sequence similarity to the variable regions of some immunoglobulin supergene family members [74–76]. Large-scale analyses have found it as the content of urinary exosomes in healthy patients [77] and detected an A1BG fragment significantly abundant in the urine of pediatric steroid-resistant nephrotic

syndrome patients [78]. NHL repeat-containing protein 3 (NHLRC3) is a protein containing NCL-1, HT2A, and Lin-41 (NHL) family repeats; not associated with any specific biological pathway, its NHL-repeat content suggests it may be involved in a variety of enzymatic processes, such as protein modification, particularly ubiquitination [76]. Literature regarding the functions or disease correlations of A1BG and NHLRC3 is scarce. Hence, our findings provide new data on these proteins. Since both A1BG and NHLRC3 are plasma proteins and an increase in A1BG levels in urine has been described in nephrotic syndrome patients, we speculate their abundance in the urine of snakebite victims may act as an early indicator of impairment in the glomerular permeability barrier [78].

Upon evaluating the biomarkers candidates already proposed for acute kidney injury and other renal diseases in our data, the results appointed retinol-binding protein (RBP4), beta-2-microglobulin (B2M), cystatin-C (CST3), hepcidin (HAMP), and fatty acid-binding protein (L-FABP) as promising candidates for kidney function surveillance in *Bothrops* envenoming. Previously, RBP4 was suggested as a prognostic marker for AKI, as its urine levels responded more sensibly than serum creatinine in monitoring AKI patients' recovery [48]. Increased levels of B2M in urine have been associated with renal tubular injury following renal allografts procedures and toxins exposure, including viper venom [49,79,80]. CST3 was identified as a candidate marker for early diagnosis and prognosis of AKI stages 2 and 3, during the 4–8-hour period post-bite in Russel's viper envenoming [22]. Like serotransferrin, hepcidin is an iron-binding protein, and its urine levels were recorded to increase in lupus nephritis [81]. Interestingly, the isoform hepcidin-25 was accounted as a prognostic marker for non-AKI outcomes in patients who went through cardiopulmonary bypass surgeries [82]. In our data, the candidate biomarker associated with the No-AKI group was calprotectin, a complex composed of the proteins S100-A8 (S100A8) and S100-A9 (S100A9). This pro-inflammatory complex presents a mediator role in inflammation and the immune system and prevents exacerbated tissue damage by scavenging oxidants [46,47]. A decrease in urine calprotectin levels was observed in kidney sclerosis in an autoimmune disease setting [83], while increased levels were found in chronic kidney disease when compared to healthy controls [82]. Their abundance in the urine of No-AKI patients may suggest a protective role in *Bothrops* snakebite envenoming. Lastly, NGAL, AHSG, and clusterin displayed no significant differential abundance in *Bothrops* snakebite-induced AKI, despite being described as promising biomarkers candidates for AKI development, even in viper envenoming [18,42,43,45]. Others, such as angiopoietin-1 (ANG1), kidney injury molecule-1 (KIM-1), and the urinary monocyte chemotactic protein-1 (MCP1) were not identified [22]. The absence of VCAM-1 and ANG-1 in our study's urine samples might be explained by the accident severity and the time point of sample collection. VCAM-1 and ANG-1 are urinary biomarker candidates that have been associated with mild cases of envenoming, and their levels typically peak within 12–16 hours after the bite [24,25]. In urine analysis, considering the time point perspective is crucial, as several biomolecules exhibit a fine-tuned behavior in response to physiological changes, and their abundance may fluctuate over short or long periods [22,24,42,48]. For instance, VCAM-1 and ANG-1, as mentioned above, show variation in their levels within a specific time frame after the envenoming. However, this study is limited to a single time point observation.

Moreover, the naturally high variability in urine poses a challenge for analyzing these samples, particularly for biomarker research [19,20,42]. Various factors are already known to influence the urinary protein landscape, such as the patient's sex and age, the timing of sample collection, the presence of comorbidities, dietary composition, the volume of venom inoculated, and even the time between the bite and hospital care [22,63,84]. Thus, the results of this work are also limited by the small size cohort of AKI patients, the few female patients observed, the presence of AKI patients with hypertension, and the analysis of only AKI at Stage 2; thus,

further studies with larger cohorts observing these aspects are needed to validate and reinforce the results presented in our study. These limitations impose still unknown effects on the data presented, especially the presence of hypertensive patients. Hypertension frequently causes renal damage, the extension of it is correlated to several factors, such as individual susceptibility, the degree of hypertension, or even if it's being properly treated or not [85]. These possible injuries might lead to a higher abundance of some proteins and create a bias for these in the data. However, hypertension is a common comorbidity in the general population and among AKI patients [86], then, we opted to include this population in our work since our proposal was to observe a more 'closer to reality' cohort when designing the study.

Regarding limitations on protein identification, we were not able to identify *Bothrops* venom-derived protein or peptide in the urine samples. This lack of identification derives from the challenges of studying a non-model organism due to fewer curated gene or transcript sequence databases [26,87,88]. In addition to this, the diversity of toxins proteoforms in snake venoms presents an additional challenge to bottom-up analysis of complex samples, such as those analyzed in this study [26,89]. Lastly, it's noteworthy that absolute quantitation through labeling methods is considered the gold standard for quantitation in proteomics due to its higher accuracy and reproducibility when compared to label-free methods [89]. We performed a label-free protocol with relative quantitation by the extracted ion chromatogram (XIC) strategy. This means our quantitative results, and the conclusions derived from them, are impacted by the limitations on the accuracy of this method's use. Still, among the label-free quantitation protocols, the approach we applied—extracted ion chromatogram (XIC)- has the greater accuracy [89]. In addition, the use of a label-free protocol presents the benefit of identifying a larger number of proteins with a wider dynamic range of detection, when compared to labeling-methods [89]. We believe the use of a label-free protocol was sufficient to achieve the intended aim of exploring the protein profile of snakebite victims who developed AKI, a still understudied topic.

## 5. Conclusions

This study explored the mechanisms leading to acute kidney injury (AKI) caused by *Bothrops* envenoming. The urinary protein profile of AKI patients revealed associations with acute phase response, endopeptidase inhibition, complement cascade, and inflammation. Specific proteins, such as serotransferrin, alpha-trypsin 1, alpha-1B-glycoprotein, and NHLRC3, were identified as differentially abundant in AKI, potentially contributing to early AKI development. On the other hand, the No-AKI group showed proteins related to hemostasis and immune response. Promising biomarker candidates for kidney damage in *Bothrops* snakebites were retinol-binding protein (RBP4), beta-2-microglobulin (B2M), cystatin-C (CST3), hepcidin (HAMP), and fatty acid-binding protein (L-FABP). In contrast, NGAL, AHSG, and CLU did not perform well as early biomarkers for AKI-induced envenoming. Future studies with larger cohorts and multiple time points are needed to validate and expand these findings. Overall, this work sheds light on the physiological disturbances and potential biomarkers associated with AKI in *Bothrops* envenoming.

## Supporting information

**S1 File. Full list of identified proteins.**
(XLSX)

**S2 File. Full list of identified proteins according to renal function status.**
(XLSX)

## Acknowledgments

For all the help during the research, we would like to thank the team at the Clinical Analysis Laboratory of the Dr. Heitor Vieira Dourado Foundation of Tropical Medicine; the Proteomics group and the Biology of Host-Pathogen Interaction Postgraduate program (PPGBIO) from Leônidas and Maria Deane Institute—Fiocruz Amazônia; the Laboratory of Hemostasis of the Hospital Foundation of Hematology and Hemotherapy of Amazonas; the Immunopathology Laboratory of the Butantan Institute, São Paulo-SP. We would also like to thank the Mass Spectrometry Platform RPT02H and the Structural and Computational Proteomics Laboratory team at Carlos Chagas Institute—Fiocruz Paraná, for all the technical and data analysis support.

## Author Contributions

**Conceptualization:** Fan Hui Wen, Ana Maria Moura-da-Silva, Jacqueline Sachett, Priscila Ferreira de Aquino, Wuelton M. Monteiro.

**Data curation:** Lisele Maria Brasileiro-Martins, Sofia Angiole Cavalcante, Marlon Dias Mariano Santos, Amanda C. Camillo-Andrade, Juliana de Saldanha da Gama Fischer, Priscila Ferreira de Aquino.

**Formal analysis:** Lisele Maria Brasileiro-Martins, Sofia Angiole Cavalcante, Alexandre Vilhena Silva-Neto, Marlon Dias Mariano Santos, Amanda C. Camillo-Andrade, Juliana de Saldanha da Gama Fischer, Priscila Ferreira de Aquino.

**Funding acquisition:** Marco Aurelio Sartim, Priscila Ferreira de Aquino, Wuelton M. Monteiro.

**Investigation:** Sofia Angiole Cavalcante, Thaís Pinto Nascimento, Amanda C. Camillo-Andrade, Juliana de Saldanha da Gama Fischer, Caroline Coelho Ferreira, Lucas Barbosa Oliveira, Marco Aurelio Sartim, Allyson Guimarães Costa, Manuela B. Pucca, Priscila Ferreira de Aquino.

**Methodology:** Sofia Angiole Cavalcante, Amanda C. Camillo-Andrade, Lucas Barbosa Oliveira, Allyson Guimarães Costa, Manuela B. Pucca, Jacqueline Sachett, Paulo Costa Carvalho, Priscila Ferreira de Aquino, Wuelton M. Monteiro.

**Project administration:** Jacqueline Sachett, Priscila Ferreira de Aquino, Wuelton M. Monteiro.

**Resources:** Paulo Costa Carvalho, Priscila Ferreira de Aquino, Wuelton M. Monteiro.

**Software:** Alexandre Vilhena Silva-Neto, Marlon Dias Mariano Santos.

**Supervision:** Jacqueline Sachett, Paulo Costa Carvalho, Priscila Ferreira de Aquino.

**Validation:** Paulo Costa Carvalho, Priscila Ferreira de Aquino.

**Visualization:** Lisele Maria Brasileiro-Martins, Thaís Pinto Nascimento.

**Writing – original draft:** Lisele Maria Brasileiro-Martins, Sofia Angiole Cavalcante, Thaís Pinto Nascimento, Alexandre Vilhena Silva-Neto.

**Writing – review & editing:** Marlon Dias Mariano Santos, Amanda C. Camillo-Andrade, Juliana de Saldanha da Gama Fischer, Caroline Coelho Ferreira, Lucas Barbosa Oliveira, Marco Aurelio Sartim, Allyson Guimarães Costa, Manuela B. Pucca, Fan Hui Wen, Ana Maria Moura-da-Silva, Jacqueline Sachett, Paulo Costa Carvalho, Priscila Ferreira de Aquino, Wuelton M. Monteiro.

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
