## [Decision Letter · Decision Letter 0]

8 Jan 2024

Dear Dr Monteiro,

Thank you very much for submitting your manuscript "Urinary Proteomics Reveals Biological Processes Related to Acute Kidney Injury in Bothrops atrox Envenomings" for consideration at PLOS Neglected Tropical Diseases. As with all papers reviewed by the journal, your manuscript was reviewed by members of the editorial board and by several independent reviewers. In light of the reviews (below this email), we would like to invite the resubmission of a significantly-revised version that takes into account the reviewers' comments. 

We cannot make any decision about publication until we have seen the revised manuscript and your response to the reviewers' comments. Your revised manuscript is also likely to be sent to reviewers for further evaluation.

Sincerely,

Wayne Hodgson

Academic Editor

José María Gutiérrez

Section Editor

Reviewer's Responses to Questions

**Key Review Criteria Required for Acceptance?**

**Methods**

-Are the objectives of the study clearly articulated with a clear testable hypothesis stated?

-Is the study design appropriate to address the stated objectives?

-Is the population clearly described and appropriate for the hypothesis being tested?

-Is the sample size sufficient to ensure adequate power to address the hypothesis being tested?

-Were correct statistical analysis used to support conclusions?

-Are there concerns about ethical or regulatory requirements being met?

Reviewer #1: -Are the objectives of the study clearly articulated with a clear testable hypothesis stated? Yes

-Is the study design appropriate to address the stated objectives? Yes

-Is the population clearly described and appropriate for the hypothesis being tested? Yes

-Is the sample size sufficient to ensure adequate power to address the hypothesis being tested? Satisfactory 

-Were correct statistical analysis used to support conclusions? Yes

-Are there concerns about ethical or regulatory requirements being met? No, ethical approval has been sorted.

Reviewer #2: Yes, the methodologies used look fine, and there are no ethical issues. However, additional details on the exclusion and inclusion criteria used are needed.

**Results**

-Does the analysis presented match the analysis plan?

-Are the results clearly and completely presented?

-Are the figures (Tables, Images) of sufficient quality for clarity?

Reviewer #1: -Does the analysis presented match the analysis plan? Yes

-Are the results clearly and completely presented? Yes

-Are the figures (Tables, Images) of sufficient quality for clarity? Yes

Reviewer #2: Yes, the data presented look good and are in line with the study plans.

**Conclusions**

-Are the conclusions supported by the data presented?

-Are the limitations of analysis clearly described?

-Do the authors discuss how these data can be helpful to advance our understanding of the topic under study?

-Is public health relevance addressed?

Reviewer #1: -Are the conclusions supported by the data presented? Yes

-Are the limitations of analysis clearly described? Yes

-Do the authors discuss how these data can be helpful to advance our understanding of the topic under study? Yes

-Is public health relevance addressed? To a certain extent.

Reviewer #2: Yes, the conclusions are fair although more details on various limitations are required.

**Editorial and Data Presentation Modifications?**

Reviewer #1: (No Response)

Reviewer #2: NA

**Summary and General Comments**

Reviewer #1: The paper describes the proteomic profile of AKI, non-AKI Bothrops envenomated victims and health control targeting to understand the mechanism of developing AKI in Bothrops envenomation. 

Major comments.

Method of creatinine and other assays have to describe under the methods.

According to AKIN criteria, how many of the patients were in stage 1, 2 and 3. That is very important to interpret the proteomics in AKI. highly recommend to present the proteomic data according to AKI stages.

What is the sample collection timing of AKI and non-AKI groups following snakebite? These data should be well presented under the results. If sample collection happened sooner after the bite in non-AKI group, some of these patient may develop AKI alone the time course. This would significantly affect the differentiation of two groups. 

Describe the other clinical features between AKI and non-AKI groups and present under the results. 

Line 266: 75% had hypertension: How authors can attribute the changes in proteomics happened to the snakebite, and not due to hypertension? What type of proteomics baseline reported due to hypertension and this has to be discussed in details in the discussion. 

Line 266: What does meant by "moderate accidents"?

Two different citation styles have been used to cite the references in the text. Please correct it according to PLoS citing style.

There are many discussion points have been included and discussed under the results sections and cited the relevant references. Since the article style has a separate section for the discussion, please move all the discussion points included in the results to under the discussion section. State only the results of THIS study under the discussion. 

Line 161: "Two microliters (2 ml) in two places. Can't understand whether this should be 2 milliliters or 2 microliters. Please correct to avoid confusion. I think thus should be 2 ml. 

Include supplementary file one as a table one in the main article. 

Minor comments:

Line 48, 49 and few more places: "bothropic" should be spelled as "Bothropic"

Reviewer #2: In this article, Monteiro and colleagues report the proteomic analysis of urine samples collected from bothrops bite patients who developed acute kidney injury and compared with snakebite patients who did not develop this condition along with healthy controls. Overall, the study has been well planned and the data aligns with the conclusions drawn. Since developing biomarkers for snakebite envenoming is key for improving the clinical management of this condition, this study is timely and will be useful to improve the understanding of bothrops envenomings. 

I suggest the authors address the following comments to improve the clarity of information presented in this article. 

1. As I stated above, more details are needed to explain the inclusion and exclusion criteria used in this study to recruit the patients. How did the authors ascertain AKI in these patients? Which methods were used? what stages of AKI were developed in these patients in line with AKIN criteria? What is the likelihood of patients having undiagnosed previous health conditions including AKI? It would be helpful to address all these points with further details. 

2. At what time points, serum creatinine was measured? I see one was done before antivenom? when was the second one? At what stage does the creatinine level get elevated to ascertain AKI in bothrops envenomings? 

3. In line 161, is it micro or millilitres? 

4. In some places, the snake genus or species names are not in italics. 

5. in line, it was stated that 55% of patients experienced moderate accidents. what does this mean? Please explain this in more detail. 

6. I would suggest the authors give more focus on the actual mass spec data obtained in this study. For example, which parameters were used to differentiate the proteins from baseline noise to full proteins/peptides? was the abundance of each protein clearly quantified? It's better to explain these details in more detail compared to the predicted results in the later sections. 

7. Would it be possible to correlate the mass spec data with specific parameters of patients? for example, male patients Vs females? different age categories? time of bite? or time to antivenom? or any of their clinical symptoms or haematological parameters? This correlation will be helpful to link these protein compositions to the actual manifestations. 

8. The limitations of this study need to include more details. The number of patients should be included in future studies as 10 patients may not be enough to draw firm conclusions. More females are needed in future to support data comparisons. Similarly, the limitations should include the time of bite and antivenom doses. 

9. I suggest the authors avoid including small paragraphs in the discussion. It's better to combine all the information relating to each theme in a single paragraph. 

10. Finally, I strongly recommend authors thoroughly proofread this article as there are some typographical and grammatical errors.

PLOS authors have the option to publish the peer review history of their article (what does this mean?). If published, this will include your full peer review and any attached files.

Reviewer #1: Yes: Kalana Maduwage

Reviewer #2: No
---

## [Decision Letter · Decision Letter 1]

14 Mar 2024

Dear Dr Monteiro,

We are pleased to inform you that your manuscript 'Urinary Proteomics Reveals Biological Processes Related to Acute Kidney Injury in Bothrops atrox Envenomings' has been provisionally accepted for publication in PLOS Neglected Tropical Diseases.

Best regards,

Wayne Hodgson

Academic Editor

José María Gutiérrez

Section Editor

Reviewer's Responses to Questions

**Key Review Criteria Required for Acceptance?**

**Methods**

-Are the objectives of the study clearly articulated with a clear testable hypothesis stated?

-Is the study design appropriate to address the stated objectives?

-Is the population clearly described and appropriate for the hypothesis being tested?

-Is the sample size sufficient to ensure adequate power to address the hypothesis being tested?

-Were correct statistical analysis used to support conclusions?

-Are there concerns about ethical or regulatory requirements being met?

Reviewer #1: -Are the objectives of the study clearly articulated with a clear testable hypothesis stated? Yes

-Is the study design appropriate to address the stated objectives? Yes

-Is the population clearly described and appropriate for the hypothesis being tested? Yes

-Is the sample size sufficient to ensure adequate power to address the hypothesis being tested? Yes

-Were correct statistical analysis used to support conclusions? Yes

-Are there concerns about ethical or regulatory requirements being met? Yes

Reviewer #2: Yes

**Results**

-Does the analysis presented match the analysis plan?

-Are the results clearly and completely presented?

-Are the figures (Tables, Images) of sufficient quality for clarity?

Reviewer #1: -Does the analysis presented match the analysis plan? Yes

-Are the results clearly and completely presented? Yes

-Are the figures (Tables, Images) of sufficient quality for clarity? Yes

Reviewer #2: Yes

**Conclusions**

-Are the conclusions supported by the data presented?

-Are the limitations of analysis clearly described?

-Do the authors discuss how these data can be helpful to advance our understanding of the topic under study?

-Is public health relevance addressed?

Reviewer #1: -Are the conclusions supported by the data presented? Yes

-Are the limitations of analysis clearly described? Yes

-Do the authors discuss how these data can be helpful to advance our understanding of the topic under study? Yes

-Is public health relevance addressed? Yes

Reviewer #2: Yes

**Editorial and Data Presentation Modifications?**

Reviewer #1: None

Reviewer #2: NA

**Summary and General Comments**

Reviewer #1: The revised version of the paper has been being edited by addressing all the comments raised by the reviewers. Hence, I recommend to accept this version after considering the editorial approval.

Reviewer #2: The authors have addressed all my comments carefully, and the manuscript looks much better.

PLOS authors have the option to publish the peer review history of their article (what does this mean?). If published, this will include your full peer review and any attached files.

Reviewer #1: **Yes: **Kalana Maduwage

Reviewer #2: **Yes: **Professor Sakthivel Vaiyapuri, University of Reading, UK

---

## [Editor Report · Acceptance letter]

21 Mar 2024

Dear Dr. Monteiro,

We are delighted to inform you that your manuscript, "Urinary Proteomics Reveals Biological Processes Related to Acute Kidney Injury in Bothrops atrox Envenomings," has been formally accepted for publication in PLOS Neglected Tropical Diseases.

Best regards,

Shaden Kamhawi

co-Editor-in-Chief

Paul Brindley

co-Editor-in-Chief
